# Contrasting Volatilomes of Livestock Dung Drive Preference of the Dung Beetle *Bubas bison* (Coleoptera: Scarabaeidae)

**DOI:** 10.3390/molecules27134152

**Published:** 2022-06-28

**Authors:** Nisansala N. Perera, Paul A. Weston, Russell A. Barrow, Leslie A. Weston, Geoff M. Gurr

**Affiliations:** 1Gulbali Institute of Agriculture, Water and Environment, Charles Sturt University, Wagga Wagga, NSW 2678, Australia; nperera@csu.edu.au (N.N.P.); pweston@csu.edu.au (P.A.W.); rubarrow@csu.edu.au (R.A.B.); leweston@csu.edu.au (L.A.W.); 2School of Agriculture, Environment and Veterinary Sciences, Charles Sturt University, Wagga Wagga, NSW 2678, Australia; 3School of Agriculture, Environment and Veterinary Sciences, Charles Sturt University, Leeds Parade, Orange, NSW 2800, Australia

**Keywords:** dung volatiles, VOCs, olfactometer, SPME, GC–MS/QToF, EAG, electroantennogram, dung beetle attraction, insect behavior

## Abstract

Volatile cues can play a significant role in the location and discrimination of food resources by insects. Dung beetles have been reported to discriminate among dung types produced by different species, thereby exhibiting behavioral preferences. However, the role of volatile organic compounds (VOCs) in dung localization and preference remains largely unexplored in dung beetles. Here we performed several studies: firstly, cage olfactometer bioassays were performed to evaluate the behavioral responses of *Bubas bison* (Coleoptera: Scarabaeidae) to VOCs emanating from fresh horse, sheep, and cattle dung; secondly, concurrent volatilome analysis was performed to characterize volatilomes of these dung types. *Bubas bison* adults exhibited greater attraction to horse dung and less attraction to cattle dung, and they preferred dung from horses fed a pasture-based diet over dung from those fed lucerne hay. Volatilomes of the corresponding dung samples from each livestock species contained a diverse group of alkanes, alkenes, alkynes, alcohols, aldehydes, ketones, esters, phenols, and sulfurous compounds, but the composition and abundance of annotated VOCs varied with dung type and livestock diet. The volatilome of horse dung was the most chemically diverse. Results from a third study evaluating electroantennogram response and supplementary olfactometry provided strong evidence that indole, butyric acid, butanone, *p*-cresol, skatole, and phenol, as well as toluene, are involved in the attraction of *B. bison* to dung, with a mixture of these components significantly more attractive than individual constituents.

## 1. Introduction

Dung beetles (mainly in the coleopteran family Scarabaeidae) are a ubiquitous group of beneficial insects that consume the dung of other animals, mainly mammals, for their survival. They play a critical role in decomposition and nutrient cycling in most terrestrial ecosystems and are important for the productivity and health of livestock [1,2,3]. Most species of dung beetles subsist on excrement from a range of animal species and are thus considered generalists [4]. The ability of dung beetles to sense and navigate to dung pats is dependent on the olfactory response to volatile organic compounds (VOCs) from dung [5,6,7]. These VOCs typically reveal the location, type, and condition of unpredictable and patchily distributed food resources for dung-feeding organisms [8,9]. Notwithstanding a capacity for generalism, most dung beetle species are preferentially attracted to fresh dung (which is likely to be less heavily utilized by earlier-arriving competitors) due to elevated levels of volatile emission compared with dried dung [10,11]. Moreover, aging and the degree of decomposition of dung affect the numbers and species of dung beetles colonizing a dung pat over time [12]. However, discrimination has also been observed based on dung type (e.g., whether produced by omnivores, herbivores, or carnivores [13,14]), with certain dung beetle species preferentially responsive to dung odors emanating from dung produced by certain species within a feeding guild [15,16]. For example, Dormont et al. (2007) observed the olfactory preference of 11 field-collected species of dung beetles to volatiles from one of four dung types offered (sheep, cattle, deer, and horse) in behavioral bioassays [17]. This revealed that dung beetles exhibit preferences toward a certain dung type, even though most of them are considered to be generalists.

Different dung types produce diverse volatilomes composed of constituents that are consistently shared across dung types as well as those that are unique to each dung type [18]. Behavioral experiments have confirmed that odors emanating from dung are involved in resource selection by dung beetles; in laboratory assays conducted by Dormont et al. [5,18], dung beetles preferentially oriented toward volatiles from the same dung type they were attracted to in the field. Several reports suggest that traps baited with compounds identified in dung headspace using gas chromatography coupled with mass spectrometry (GC–MS) [7,18,19,20] were attractive to dung beetles [6,18,19,21]. Recently, a study confirmed the capacity of cattle dung VOCs to alter the composition of dung-inhabiting insect assemblages [22]. Further evidence for the roles played by VOCs in the behavior of dung-inhabiting insects comes from the coevolutionary phenomenon of deception of these species by dung-mimicking odoriferous plants to facilitate their seed dispersal [23,24]. However, studies to more fully characterize livestock dung volatilomes remain limited. Against this research background conducted chiefly in a European setting [6,8,19,25,26], we sought to develop a more complete understanding of dung volatilomes that utilized recent advances in GC–MS to characterize livestock dung from an Australian perspective. Our intention was to provide more robust empirical evidence for use of VOCs by dung beetles when discriminating among dung resources through behavioral assays, volatile analysis, and electroantennography for the first time. Other aspects of our methodology enabled a more detailed analysis of the olfactory ecology of dung beetles, namely (1) the use of solvent-free solid-phase microextraction (SPME) for the collection of dung VOCs, which provides greater trapping efficiency and sensitivity than other methods [27]; (2) using a non-targeted metabolomics approach employing gas chromatography coupled to quadrupole time-of-flight mass spectrometry (GC–MS/ QToF) to enable accurate mass screening of volatilomes, and (3) using dung from an attractive animal source (horse) fed two discrete diets, a novel contribution of our study that allows for increased resolution of behavioral assays.

Our study species, *Bubas bison* (Coleoptera: Scarabaeidae: Scarabaeinae) was first introduced to Australia in 1983 [28] to supplement the activity of native dung beetles which are adapted to the dry pelletized dung of native Australian marsupials and less well adapted to cope with large volumes of moist dung from introduced livestock, such as cattle. *Bubas bison* is now widely established on the Australian mainland and is one of the few winter-active species present in the country. We hypothesized that *B. bison* discriminates among dung from different livestock species and that VOCs in the dung volatilomes drive this behavior. Accordingly, the specific objectives of this study were to (1) assess the discrimination of adult *B. bison* to odors emanating from horse, cattle, and sheep dung using laboratory olfactometer bioassays; (2) identify VOCs present in the volatilomes of these dung types; (3) determine the antennal responsiveness of *B. bison* to a mix of selected chemical compounds that characterize attractive dung; and (4) observe the orientation responses of *B. bison* to single compounds or blends of selected compounds detected in the volatilome.

## 2. Results

### 2.1. Behavioral Assay to Dung VOCs

Validation tests with our cage bioassay revealed no observed bias due to the olfactometer design (*p* = 0.6355) (Figure 1a). Significantly more adult *B. bison* orientated to horse dung than cattle dung (*p* = 0.0256) (Figure 1b) and to sheep dung than cattle dung (*p* = 0.0425) (Figure 1c). However, *B. bison* did not show a statistically significant preference for horse dung over sheep dung (*p* = 0.0742) (Figure 1d), although the beetles showed a numerical bias toward horse dung. In addition, dung from pasture-fed horses was significantly more attractive than that of lucerne hay-fed horses (*p* = 0.0161) (Figure 2).

### 2.2. Volatilome Profiling

#### 2.2.1. Horse, Cattle, and Sheep Dung Volatilomes (from Pasture-Fed Animals)

Dung volatilomes from horses, cattle, and sheep were distinctive and contained a total of 127 VOCs and included compounds containing the following functional groups: alkanes, alkenes, alkynes, alcohols, aldehydes, ketones, carboxylic acids, esters, phenols, amines, ethers, oximes, sulfides, disulfides, and furans across all three dung types (Figure 3, Appendix A). The numbers of volatile compounds identified in horse, sheep, and cattle dung were 75, 85, and 63, respectively. Twenty-nine compounds were common to all three dung types, and 18, 28, and 14 compounds were unique to horse, sheep, and cattle dung, respectively (Figure 4). Overall, the horse dung volatilome had a wider range of VOCs with respect to the class of compounds (Appendix A) than cattle and sheep volatilomes. MZmine detected a total of 73 features for statistical analysis via Metaboanalyst 5.0. Score plots from PLS-DA showed distinct separation of the three dung volatilomes (Figure 5). The first principal component (PC 1), accounting for 36.3% of the total variance, described the variation of cattle dung from horse + sheep dung volatilomes, while PC 2, which explained 33.5% of the total variance, corresponded to differentiation among the three dung types. Hierarchical clustering based on ion intensities clustered the three dung types into two main VOC groups—cattle + sheep vs. horse (Figure 6)—suggesting discrimination between ruminant dung and non-ruminant dung. The accompanying heatmap illustrates the differences in key constituents among the three dung volatilomes. Most constituents, representing a broad range of chemical classes, were found in horse dung at relatively high levels compared with the other two dung types analyzed. These VOCs included 5-nonanone, 2-heptanone, toluene, skatole, 2-octanone, *p*-xylene, 2-nonanone, indole, 3-nonyne, phenol, *α*-phellandrene, and 3-ethylbenzaldehyde. The terpenoids *α*-pinene, *β*-pinene, borneol, limonene, and camphene were observed in the highest abundance in cattle dung, whereas 1,1,3-trimethyl-2-cyclohexanone, cubenol, heptane, allyl isothiocyanate, propyl 2-methylbutanoate, and (*Z*)-7-hexadecene were distinctly elevated in sheep dung.

#### 2.2.2. Horse Dung from Animals Fed on Lucerne-Hay and Pasture

Volatile profiles of dung from horses fed the two diets varied substantially in terms of VOC composition and abundance (Figure 7). The numbers of VOCs annotated from horse dung produced by lucerne hay- and pasture-fed animals were 53 and 40, respectively. Of those, 14 constituents were found in both dung types, while 39 and 26 constituents, respectively, were found only in lucerne hay and pasture-fed dung volatilomes (Appendix A). Pasture-fed horse dung had a greater abundance of skatole, eucalyptol, indole, *p*-cresol, *p*-cymene, *p*-ethylphenol, phenol, 3-nonyne, 2-heptanone, dimethyl disulfide, dimethyl trisulphide, and toluene. The volatilome of dung from lucerne hay-fed horses contained predominantly alkanes, terpene alkenes (e.g., limonene, *α*-pinene, *β*-pinene, aromandendrene, and camphene), and aromatic hydrocarbons (Appendix A). Interestingly, phenol- and sulfur-based compounds as well as indole and skatole were completely absent in the lucerne hay-fed horse dung volatilome (Appendix A).

### 2.3. Electroantennogram Responses and Behavioral Assay with Chemical Compounds

The antennae of *B. bison* females responded significantly to the formulated six-compound mixture (*p* = 0.0032, *n* = 7) compared with the control. EAG responses ranged in magnitude from 0.9 to 3.6 mV (Figure 8).

Behavioral evaluation of individual compounds frequently present in livestock dung using the same olfactometer assay showed that more beetles were responsive to *p*-cresol, skatole, butyric acid, butanone, and phenol when compared with the control, but the response was significant only for *p*-cresol (*p* = 0.031) (Figure 9). However, when presented as a mixture, these compounds elicited a significant response from *B. bison* adults compared with the control (*p* = 0.0078) (Figure 10a) but a significantly weaker response compared with fresh horse dung (*p* = 0.0078) (Figure 10b). In addition, lucerne hay-fed horse dung spiked with a mix of toluene, *p*-cresol, phenol, and skatole showed an enhanced attractancy for *B. bison* by ~64% (Appendix A).

## 3. Discussion

Supporting our initial experimental hypotheses, adult *B. bison* beetles exhibited a significant orientation preference toward horse dung volatiles in contrast to cattle or sheep dung volatiles. The cage assays devised in our experimentation removed any potential visual cues for beetles, and subsequent chemical analysis of dung showed markedly contrasting volatilomes; thus, discrimination among livestock dung VOCs was evident. In the cage assay, beetles showed a distinct preference for horse dung VOCs over cattle dung VOCs and a clear tendency to choose VOCs from horse dung over those from sheep dung. Therefore, we established that for *B. bison,* horse dung was the most attractive, and cattle dung was the least attractive among these three dung types. In a previous study and in agreement with our findings, the congeneric *B. bubalus* exhibited greater attractancy to volatiles from horse dung than those of cattle dung in a laboratory olfactometer bioassay [5]. Further testing within our system showed that olfactory choices made by *B. bison* can be influenced by the diet of the dung producer. We demonstrated that *B. bison* adults responded preferentially to dung VOCs from horses fed on pasture compared to those fed on lucerne hay. Even though these dung types differ in moisture content, this property was shown not to be significant in explaining olfactory responses by dung beetles [13,29], supporting the conclusion that differences in the volatilomes of horse dung from animals fed different diets account for their differential attractancy. 

The dung types tested in our bioassays possessed unique volatilomes varying in both abundance and composition of VOCs. Some of the VOCs detected were previously reported to occur in dung headspace [7,18,19,20,21,30], and a number were found to be electroantennographically active (EAG-active) for certain dung beetle species, especially as constituents of pheromones [31,32,33,34,35,36]. However, the non-targeted analysis of the present study provides a more comprehensive picture of livestock dung volatilomes in terms of both the dung type and the host diet. To our knowledge, this is the only study utilizing the SPME–GC–MS/QToF system to study dung volatilomes, thus providing more comprehensive insights into chemical constituents of dung volatile profiles. Therefore, the result of the present study expands the current knowledge on VOCs present in livestock dung headspace and VOCs likely to be involved in trophic interactions in dung beetles.

With regard to dung type, the volatilome of horse dung had elevated levels of many VOCs compared with sheep and cattle dung. Specifically, we found higher levels of indole, skatole, butanone, phenol, toluene, 2-heptanone, and *β*-citronellene in the horse dung volatilome, along with moderate levels of *p*-cresol, similar to that of sheep dung. Greater attractiveness of horse dung to *B. bison* may be associated with enhanced levels of these metabolites. In terms of diet, the dung volatilome of pasture-fed horses predominantly had relatively greater amounts of indole, *p*-cresol, skatole, phenol, *p*-ethylphenol, eucalyptol, *β*-citronellene, 2-heptanone, toluene, dimethyl disulfide (DMDS), and dimethyl trisulfide (DMTS). The significant olfactory responses of *B. bison* to horse dung VOCs over sheep and cattle dung VOCs and to pasture-fed horse dung VOCs over lucerne hay-fed horse dung VOCs provide strong evidence for the role of the afore-mentioned metabolites in dung attractiveness. Indole, *p*-cresol, skatole, butyric acid, butanone, phenol, and *p*-ethyl phenol have been reported in dung smell-mimicking seeds [24] and flowers [37,38]. It is suggested that flowers producing skatole and indole may have a pre-existing bias with their beetle pollinators, explaining possible events of convergent evolution [37,39]. Moreover, skatole and indole were identified as EAD-active components in the male abdominal secretions of the African dung beetles *Kheper nigroaeneus*, *K. lamarcki*, *K. bonellii,* and *K. subaeneus* (Coleoptera: Scarabaeidae: Scarabaeinae) [32,34,35,36]. In addition, phenol and *p*-cresol have been reported as olfactory cues for *Onthophagus binodis* (Coleoptera: Scarabaeidae: Scarabaeinae), which stimulate their food searching behavior [40]. DMTS has been found to be associated with late dung-inhabiting beetles [22] and the presence of both DMDS and DMTS in chemical baits strongly attracts carrion beetles [41]. In addition, DMTS emissions from vertebrate cadavers were found to be both EAG-active and field attractive to the copro-necrophagous dung beetle *Anoplotrupes stercorosus* (Coleoptera: Geotrupidae: Geotrupinae) [26].

The results showed that the presence of certain VOCs, such as *α*-pinene, *β*-pinene, and limonene, characterize less-preferred dung types. The volatilomes of dung least preferred by *B. bison* in this study (from cattle and lucerne hay-fed horses) showed greater abundances of *α*-pinene, *β*-pinene, and limonene compared with other dung types. Dung from herbivorous animals that contains *α*-pinene has been found to be unattractive for the dung beetle *Saphobius edwardsi* [7]. On the other hand, horse dung produced by animals fed lucerne hay was free of any sulfur-containing or phenolic compounds, which can be highly odorous even at low concentrations [42], as well as indole and skatole. This could explain the comparatively lower attractiveness of cattle dung and lucerne hay-fed horse dung for *B. bison*. The intermediate attractiveness of sheep dung might be explained by the similar abundance of both *p*-cresol and butanone in both sheep and horse dung; these constituents have previously been reported to be attractive for dung beetles in the genus *Kheper* as pheromone components [34,35] and as constituents that initiated food searching behavior in the dung beetle *Geotrupes auratus* [21]. Although eucalyptol and limonene reportedly elicit antennal responses in another species of dung beetle (*O. binodis*) [40], no apparent role of these VOCs in attracting *B. bison* was noted. However, the degree of attractiveness of odor sources for beetles can be influenced by the differential abundance and/or ratios of components as well as the concentration of VOCs regardless of the number of components in a blend [43,44]. Further work focused on quantification of dung VOCs and dose-dependent EAG studies are required to assess the importance of compound abundance and ratios involved in dung beetle attraction and discrimination.

EAG recordings revealed that the six-compound mix formulated from VOCs commonly present in livestock dung elicited significant antennal responses in adult beetles, indicating the presence of chemosensory receptors for these constituents in *B. bison*, which could either be specialist or generalist receptors. To the best of our knowledge, this is the first report documenting either behavioral or antennal responses of *B. bison* to any VOC. The compounds tested were present in dung samples used in the current study and have been used in field experiments to evaluate dung beetle attractancy in European settings [6,19]. In our samples, butyric acid was detected at trace levels below the user-defined threshold, and therefore, it was not included in the statistical analysis. However, given the reported importance of butyric acid as a dung VOC and a semiochemical [34,35], we included it in our screening for EAG activity and single-compound olfactometry assays. The appreciable abundance of indole, *p*-cresol, skatole, butanone, and phenol in dung volatilomes, especially in pasture-fed horse dung, provide evidence for the role of these VOCs in dung localization by *B. bison* and likely accounts for the attraction to horse dung. Our results from the single compound bioassays suggest the significance of *p*-cresol compared with other compounds tested. Enhanced attractancy for adult *B. bison* in the cage bioassay was observed when beetles were offered the six-compound mixture. We speculate that together, these dung metabolites have a synergistic effect on beetle attraction and that *B. bison* uses a blend of compounds rather than single compounds as olfactory cues to locate dung. Here, one or more constituents in the odor bouquet can act as ‘background’ volatiles, improving the attractancy for beetles [45,46,47]. Supporting this phenomenon, the bioassay with spiked horse dung showed the ability of augmented abundance of toluene, *p*-cresol, phenol, and skatole to enhance the attractancy of horse dung. Unfortunately, the limited availability of live dung beetles at the time of experimentation constrained the number of replicates conducted and did not allow for robust statistical testing of this notion.

In conclusion, the combined results of EAG testing and bioassays with purified compounds confirmed the bioactivity of the six-compound mix, regardless of the exact compound proportions. It is possible that certain individual metabolites could have greater influence than others present in the mix, which could potentially alter the degree of attractancy, but the quantification of VOCs and the behavioral responses they elicit is required to resolve this. To date, field traps baited with dung have outperformed chemical baits, regardless of the constituents in the mix and their ratios [6,19]. However, numerous VOCs from dung volatilomes have yet to be tested for attractancy in the field. Our results from EAG and supplementary olfactometer studies suggest the potential contribution of indole, butyric acid, butanone, *p*-cresol, skatole, and phenol, as well as toluene, to the attractancy of *B. bison* to horse dung. Additionally, *p*-ethylphenol, eucalyptol, 2-heptanone, DMDS, and DMTS have been suggested as metabolites that may alter the attractancy. The results from this study will benefit the development of a blend of chemical compounds that may act as attractants for dung beetles under field conditions. This will increase the consistency and the convenience of dung beetle trapping compared with what is currently possible using dung bait.

## 4. Materials and Methods

### 4.1. Beetle and Dung Collection

Adult *B. bison* were collected from the field in the vicinity of the towns of Wagga Wagga and Tarcutta in New South Wales, Australia, between 15 and 22 July 2020. Beetles were maintained in 20 L containers filled with moistened sand and vermiculite (1:2) as a tunneling medium until use in experiments. Fresh dung was obtained from horses, sheep, and cattle feeding on pasture [mainly rye grass (*Lolium perenne* L.) and clover (*Trifolium subterraneum* L.)] and used for the olfactometry study related to different dung types; a later collection of dung from horses feeding on two distinct diets [pasture and lucerne hay (*Medicago sativa* L.)] was performed and used to evaluate the effect of diet on dung attractancy. All dung was collected at the Charles Sturt University farms at Wagga Wagga between July and October 2020. Care was taken to collect several individual fresh droppings from multiple animals that had not been treated with any anti-parasitic drugs for at least 6 weeks prior to dung collection. Dung that appeared moist and glossy was considered fresh and collected before apparent colonization by any insects. Within one hour of collection, dung was homogenized and used in bioassays, followed by volatile analysis on the same day. 

### 4.2. Olfactory Responses of B. bison to Dung Volatiles

Olfactory responses of *B. bison* to volatiles emitted from dung were assessed with a two-choice, still-air bioassay arena. Preliminary observations suggested that beetles prefer a spacious arena in which they can fly. Considering this, the olfactometer consisted of a cuboidal cage (45 × 45 × 45 cm) with screen sides and top that permitted air movement through the arena and a closable opening in one wall that permitted access to the inside (Figure 11). A false floor with two holes (2.3 cm in diameter) positioned 30 cm apart in diagonally opposed corners was placed 8 cm above the bottom to conceal the dung from the beetles’ view while allowing odors to escape and to provide a cleanable surface for beetle locomotion. Odor sources were fresh dung samples of 50 g. These were presented in plastic vials (250 mL capacity) beneath the holes in the false floor, and beetles responding to the treatment were trapped therein. The assays were conducted in a glass house under natural light conditions; the temperature fluctuated diurnally between 15 and 22 °C during the assays, which were the approximate minimum and maximum ambient temperatures at the time of the experiment. 

A total of 48 female and 48 male *B. bison* were tested in two successive rounds, and in each of eight replicates that were carried out simultaneously (16 replicates in total per assay). Insects were starved for 12 h prior to testing. Assays were started at 1500 h because of the documented evening and night activity of *B. bison* [48,49,50]. To initiate the assay, three pairs of beetles were released into the center of the arena at the release point. The openings in the false floor were covered during the first hour, which allowed beetles to acclimate to the arena and prevented accidental captures in the treatment containers. Validation of the olfactometer design was achieved by first exposing adult *B. bison* to two empty treatment containers to ascertain possible positional bias. To assess the attractiveness of different livestock dung, testing proceeded by releasing all beetles into bioassay cages in three successive assays, giving them a choice of (1) cattle vs. horse dung, (2) horse vs. sheep dung, and (3) cattle vs. sheep dung. A later round of experiments was conducted to assess the attractiveness of fresh dung collected from horses fed lucerne hay vs. those grazing on green pasture (predominantly subterranean clover and rye grass). The location of treatment containers was randomized in each cage to remove positional bias. Each round of experiments was conducted until all test beetles had reached one of the two containers, typically 12–18 h. Cages were rotated 90° during each trial at ~4 h intervals to minimize the effects of potential spatial variability in environmental factors (e.g., ambient light gradients, temperature, etc.). The number of beetles trapped in each container was recorded. Following each run, the cage floors were wiped with 70% ethanol.

Responses of *B. bison* were statistically tested using the non-parametric Wilcoxon’s signed-rank test (*p* < 0.05) [51] with Statistix 10 software (Analytical Software, Tallahassee, FL, USA). Dead beetles, which were few in number, were excluded from the analysis.

### 4.3. Volatilome Analysis

#### 4.3.1. Headspace Collection of Dung VOCs

Dung headspace VOCs were collected using solid-phase micro extraction (SPME). Fibers coated with polydimethylsiloxane/divinylbenzene (PDMS/DVB) (Agilent Technologies, CA, USA) were pre-conditioned at 270 °C for one hour, in accordance with the manufacturer’s recommendation, before use. Dung (0.7 g) was placed in 10 mL glass SPME sampling vials (Agilent Technologies, CA, USA) and incubated at 30 °C for 5 min prior to collection using an AOC 5000 auto-sampler (Shimadzu, Japan). Volatiles were collected by exposing 1 cm of the SPME fiber to the sample headspace for 30 min.

#### 4.3.2. GC–MS/QToF Analysis of Dung Volatilomes

Separation of volatile compounds was carried out using GC–MS/QToF (7890A GC; 7200 QToF; Agilent Technologies, CA, USA). Loaded SPME fibers were desorbed in the injection port at 250 °C for 1 min, and subsequent separation was achieved with an HP-5MS capillary column (30 m × 0.25 mm I.D., 0.25 μm film thickness: Agilent Technologies, CA, USA) using helium as the carrier gas at a flow rate of 1.5 mL min^−1^. The oven temperature was maintained at 40 °C for 2 min and then programmed to rise from 40 to 230 °C at a rate of 4 °C min^−1^ and then from 230 to 260 °C at a rate of 10 °C min^−1^. Mass spectra were recorded after exposing the effluent from the GC to electron ionization (EI) at 70 eV; the mass range collected was 50 to 500 *m*/*z*.

Volatiles were identified by comparing their mass spectra to reference compounds in the National Institute of Standards and Technology (NIST) mass spectrometry library and authentic standards where available. Analytical grade standards were purchased from Sigma-Aldrich, Australia. Laboratory-grade deionized water was used to dilute standards, and solutions were heated at 100 °C when needed to dissolve solid compounds. The consistency of the performance of the SPME fiber and GC–MS system was assessed by periodically injecting a test mix of *p*-cresol, phenol, and toluene at 10 ppm.

#### 4.3.3. Identification of VOCs

Spectral data were processed initially using Mass Hunter Qualitative Analysis software (Version 7.0 B). Compound identification was achieved by comparing mass spectra with entries in the NIST database (version 2.3, 2017) and known authentic standards. When standards were not available, linear retention indices were calculated using an *n*-alkane series (C_8_–C_20_) run under the same chromatographic conditions to confirm compound identity [52]. Known artifacts (e.g., siloxanes) were excluded from the identified compound list. This Mass Hunter-produced initial compound list was used to compare the presence of VOCs across different dung volatilomes by comparing retention indices and their mass spectra (Appendix A). MZmine version 2.53 [53,54] and Metaboanalyst 5.0 were used to process high-resolution molecular profile data for further analysis. The data analysis workflow consisted of sequential peak detection, deconvolution, alignment, gap-filling, and statistical analysis [53,55,56]. First, raw GC–MS/QToF data files were converted from Agilent proprietary “.d” format to “mzData” format using Agilent Mass Hunter Qualitative Analysis 7.0 and then imported into MZmine for pre-processing. Then, peak/mass detection was performed to determine the *m*/*z* for each scan based on a user-specified noise level which generated a mass list. After this centroiding process, chromatograms were constructed for each *m*/*z* value along the entire chromatogram span using the ADAP chromatogram builder. Chromatograms were smoothed to remove high-frequency noise and then deconvoluted into individual chromatographic peaks using the Wavelets (ADAP) algorithm. After individual features had been detected, spectral deconvolution was performed by applying hierarchical clustering [57,58], which yielded fragmentation spectra. Next, feature-based alignment was conducted to generate a single data matrix that contained all features across all samples in a set. Finally, the identification of features was accomplished by comparing mass spectra with the NIST’20 database and confirmed by standards when available. On the basis of this MZmine-produced aligned feature list, multivariate statistical analysis was conducted, which included partial linear discriminant analysis (PLS-DA) and hierarchical cluster analysis. PLS-DA was performed to detect maximal differences among horse, sheep, and cattle dung volatilomes. The models did not show overfitting and were found to have a good predictive ability for the derived classification (with 10-fold cross-validation). The MZmine-produced feature list contained fewer constituents compared with the initial feature list due to the exclusion of some low-abundance constituents on the basis of user-defined parameters during the analysis.

### 4.4. Electroantennogram Recordings of B. bison

Indole, *p*-cresol, skatole, butyric acid, butanone, and phenol were selected to conduct initial electroantennography tests with female *B. bison* using the Syntech EAG system (Stimulus controller CS-55, Ockenfels Syntech GmbH, Kirchzarten, Germany). These VOCs were found across our dung samples during the GC–MS analysis (although butyric acid was found only at trace levels) and are known to attract other dung beetle species in the field [6,19,59]. Freshly excised antennae from female *B. bison* were mounted between two glass electrodes (i.d. = 1.15 mm) filled with conductive gel (Spectra 360, Parker Laboratories, Fairfield, NJ, USA). Silver wire (0.35 mm diameter) was used to make electrical contact between the electrode and the amplifier. In preliminary trials, a concentration of test compounds at 10 ppm (diluted in a water matrix) was found to be the optimal concentration to stimulate the antenna without causing saturation. The odor stimuli were prepared by loading 10 µL of each stimulus solution onto a small piece of folded filter paper strip (1 × 5 cm), which was then placed inside a glass Pasteur pipette. The tip of the pipette was inserted into a small hole in the mixing tube, where charcoal purified-humidified air was blown onto the antennal preparation. Each antenna was exposed to 0.2 s stimulus puffs (200 mL/min) delivered by the stimulus controller. Responses were measured as the maximum amplitude of depolarization (mV) between the tip and the base of the antenna. EAG responses of adult *B. bison* to the six-compound mix and negative control (*n* = 7) were recorded using a data acquisition controller (IDAC-2, Ockenfels Syntech GmbH, Kirchzarten, Germany). After each stimulus application, the antenna was flushed for 5 min with humidified air to allow for recovery. After the recovery period, the antenna was exposed to two control puffs (with water applied to filter paper in the stimulus pipette) to verify that the antennae had fully recovered. A short strand of human hair was inserted in between the outer and middle lamellae to ensure maximum exposure of olfactory receptors to the stimulus air stream during the recording. For the comparison of EAG data, a two-sample *t*-test for unequal variance was performed (*p*   <  0.05) for the control and the six-compound mix using Statistix 10 software (Analytical Software, Tallahassee, FL, USA).

### 4.5. Olfactory Responses of B. bison to Selected VOCs and Dung Spiked with Selected VOCs

The attractiveness of indole, *p*-cresol, skatole, butyric acid, butanone, and phenol (the same compounds used in the EAG analysis) was evaluated individually vs. a water control and as a mixture vs. horse dung in the same bioassay (eight replicates each). All chemicals were dissolved in a water matrix at 10 ppm (Appendix A), and 100 µL of individual compounds or the six-compound mixture was used as the bait. Bait solutions were applied to a piece of filter paper that adhered to the wall of the container following wetting with the bait solution.

On the basis of visual inspection of volatile profiles of dung from horses fed lucerne hay and pasture (Figure 7), toluene, *p*-cresol, phenol, and skatole were selected to perform a preliminary experiment to see whether these constituents enhance the attractancy of horse dung from animals fed on lucerne hay. For this experiment, beetles were offered horse dung and spiked horse dung as treatments in the cage olfactometer bioassay. Spiked horse dung was prepared by applying 100 µL of a 10 ppm matrix of each compound to a piece of filter paper (2 × 7 cm) that adhered to the inner wall of the treatment vial containing 50 g of horse dung. Results from this preliminary experiment were not statistically tested because the number of replicates was insufficient, as beetles were at the end of their seasonal availability.

## Figures and Tables

**Figure 1 molecules-27-04152-f001:**
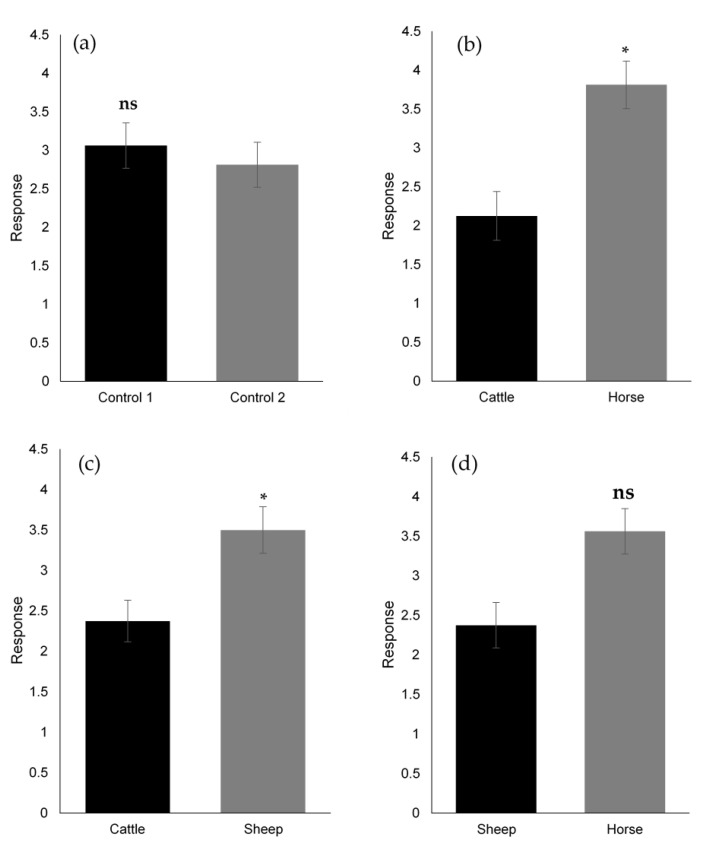
Olfactory responses of *B. bison* to fresh horse, sheep, and cattle dung (*n* = 96 beetles). (**a**) Assay validation by control comparison: (**b**) cattle vs. horse dung, (**c**) cattle vs. sheep dung, (**d**) sheep vs. horse dung (refer to 4.1 and 4.2 for experimental design and procedure. *, *p* < 0.05 and ns, not significant, as determined by Wilcoxon’s signed-rank test).

**Figure 2 molecules-27-04152-f002:**
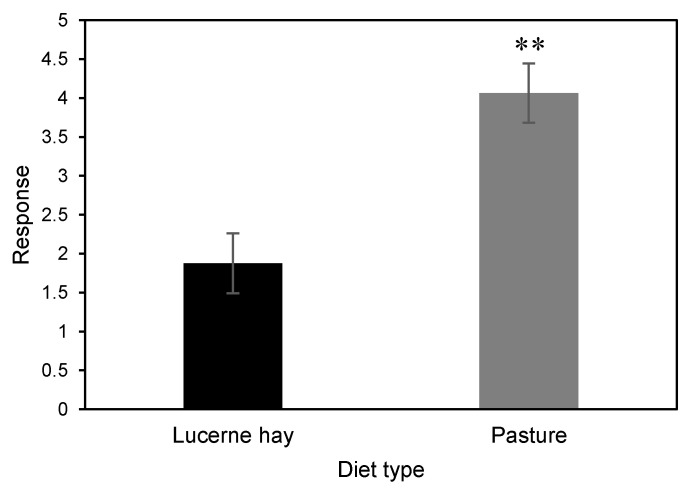
Olfactory responses of *B. bison* to fresh dung collected from horses fed on lucerne hay and fresh pasture (*n* = 96 beetles) (**, *p* < 0.01 as determined by Wilcoxon’s signed-rank test).

**Figure 3 molecules-27-04152-f003:**
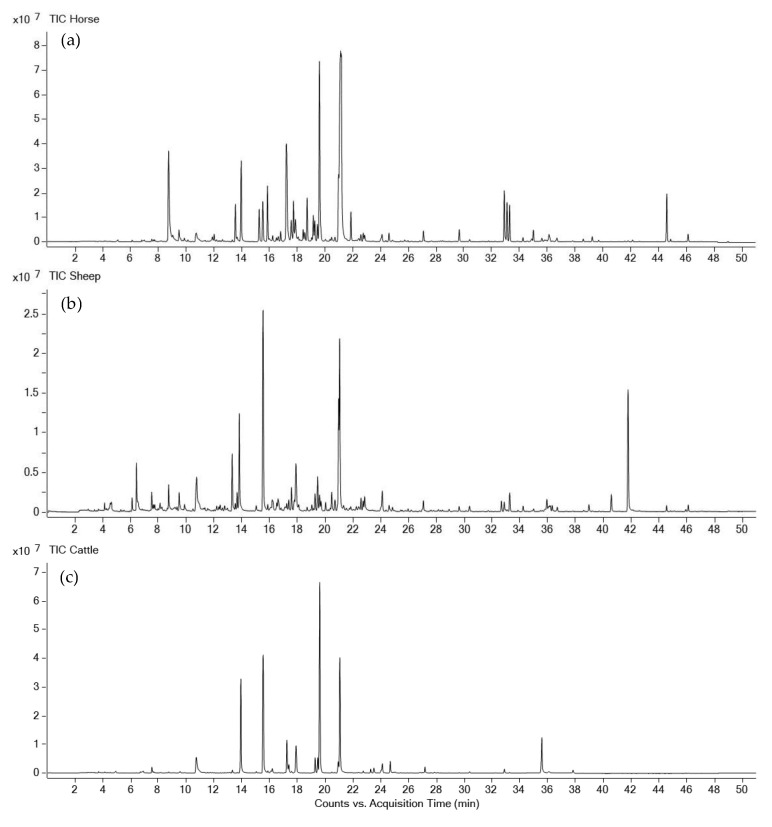
Representative gas chromatography/mass spectrometry total ion chromatograms of (**a**) horse dung, (**b**) sheep dung, and (**c**) cattle dung volatilomes.

**Figure 4 molecules-27-04152-f004:**
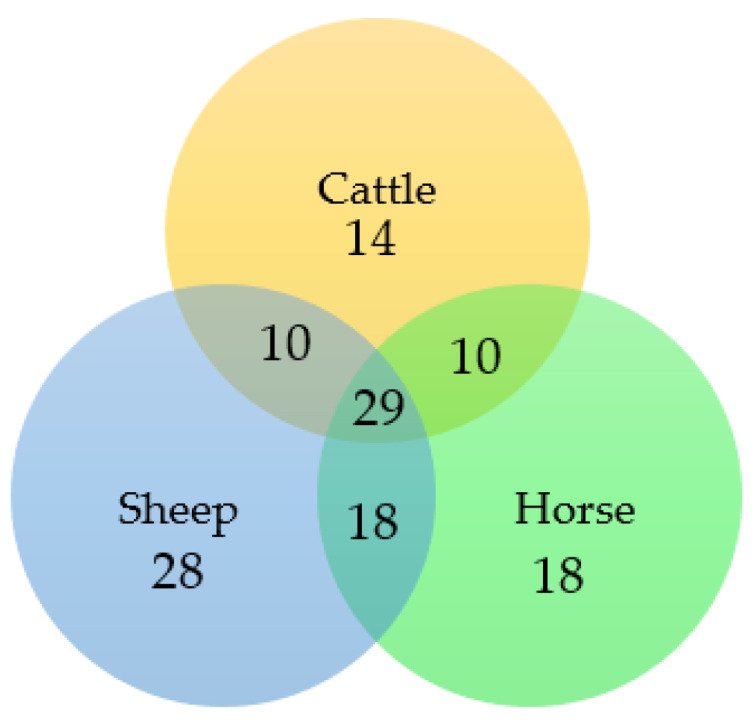
The number of molecular features (metabolites) present in horse, cattle, and sheep dung (based on the Mass Hunter-produced feature list).

**Figure 5 molecules-27-04152-f005:**
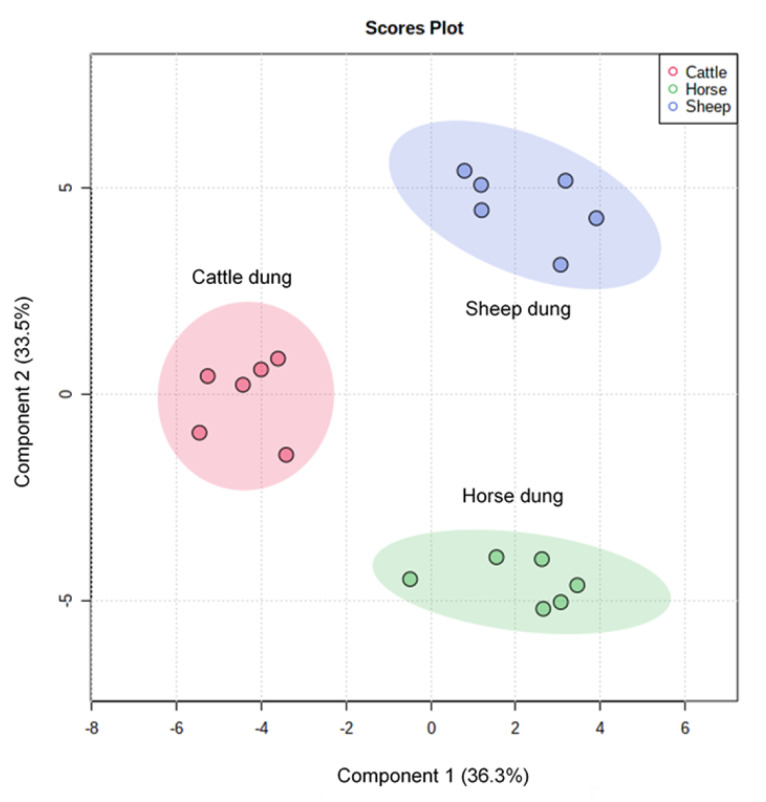
Two-dimensional scores plot from PLS-DA showing the discrimination of horse, cattle, and sheep dung based on their respective GC–MS/QToF profiles. The first and second PCs accounted for 36.3% and 33.5% of the total variation, respectively. Each point in the plot represents a replication.

**Figure 6 molecules-27-04152-f006:**
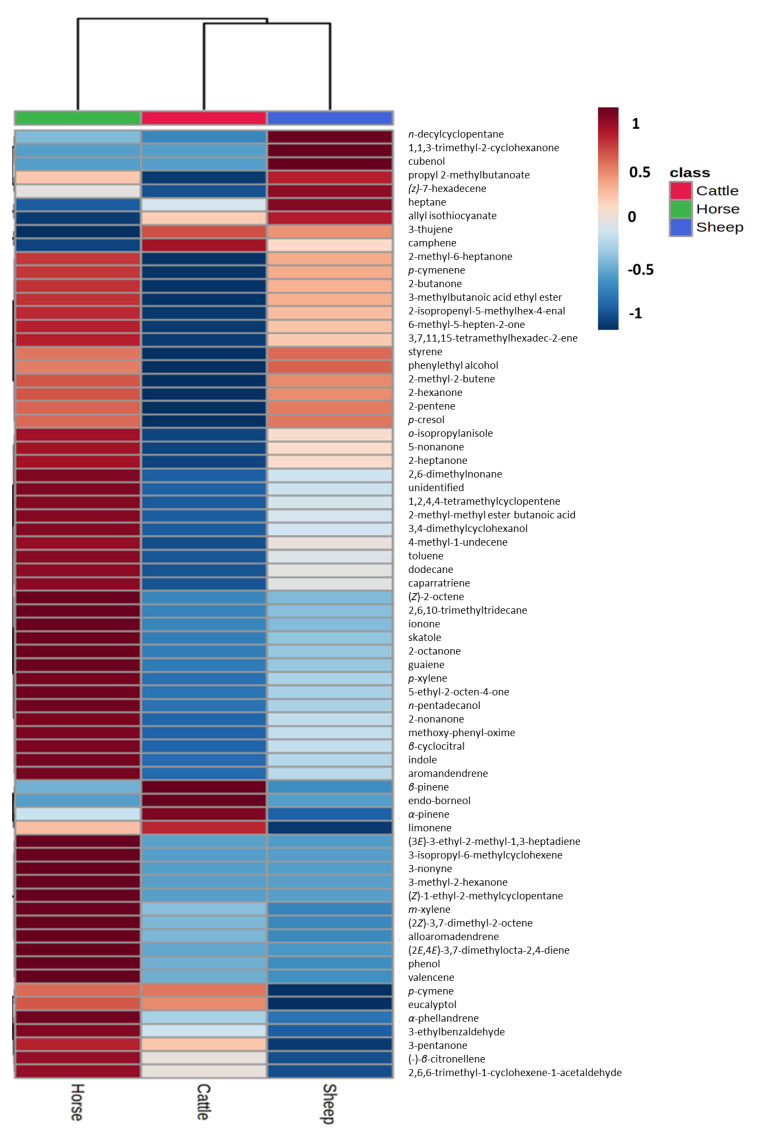
Hierarchical clustering of volatilomes of cattle, horse, and sheep dung based on the relative abundance of compounds identified in their respective headspaces (based on MZmine-produced feature list for statistical analysis).

**Figure 7 molecules-27-04152-f007:**
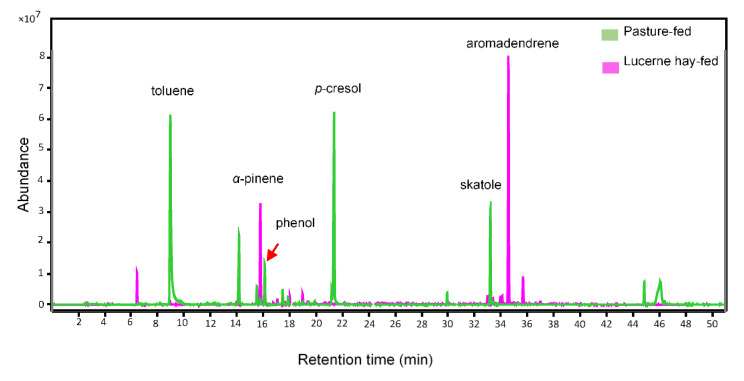
Representative gas chromatography/mass spectrometry total ion chromatograms of horse dung from animals fed on pasture and lucerne hay.

**Figure 8 molecules-27-04152-f008:**
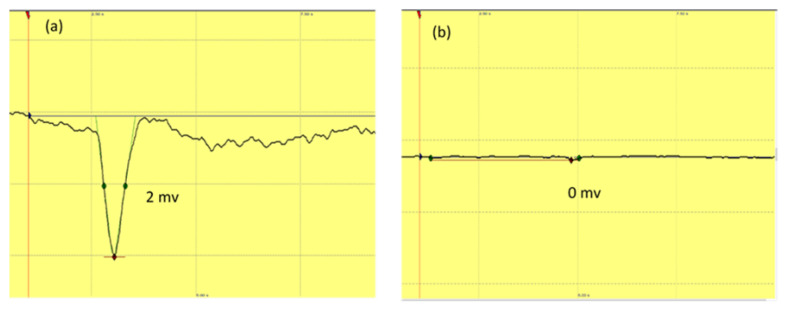
Representative EAG responses of female *B. bison* antennae to (**a**) six-compound mix (10 ppm) and (**b**) control. The stimulus was applied as a puff for 0.2 s using the SYNTEC stimulus controller (*n* = 7 beetles).

**Figure 9 molecules-27-04152-f009:**
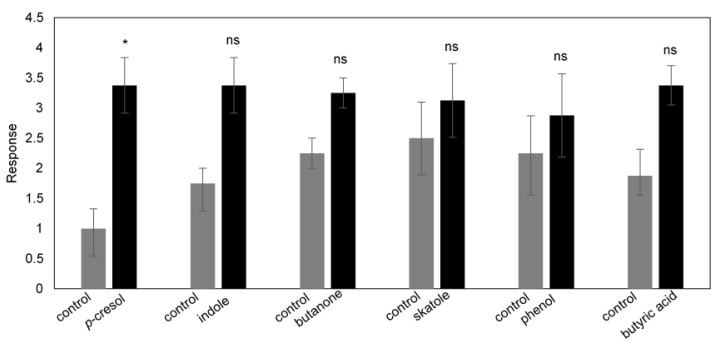
Olfactory response of *B. bison* to individual compounds (*n* = 48 beetles; *, *p* < 0.05 and ns, not significant, as determined by Wilcoxon’s signed-rank test).

**Figure 10 molecules-27-04152-f010:**
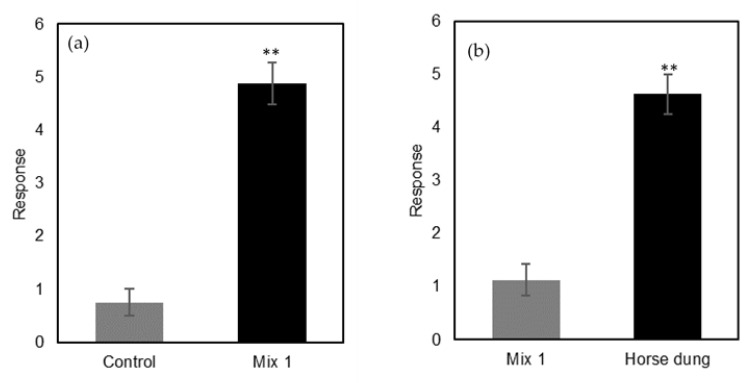
Olfactory responses of *B. bison* to (**a**) six-compound mixture (*p*-cresol, indole, butanone, skatole, phenol, and butyric acid) vs. control, and (**b**) six-compound mixture vs. fresh horse dung (*n* = 48 beetles; **, *p* < 0.01, as determined by Wilcoxon’s signed-rank test).

**Figure 11 molecules-27-04152-f011:**
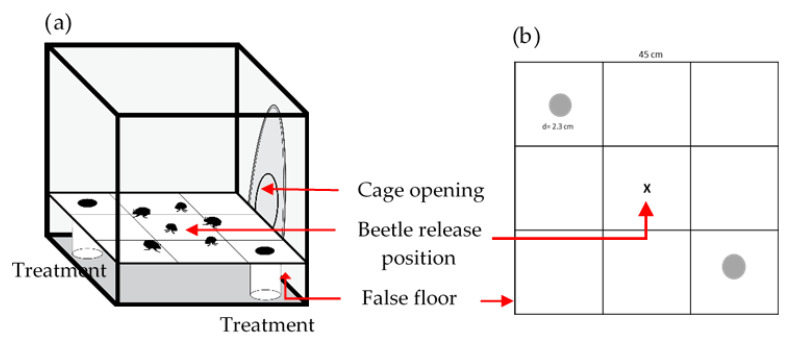
(**a**) Diagram of the cage olfactometer. Beetles were released at the central ‘X’. (**b**) False floor that provided a uniform surface for the beetles to walk upon and prevented contact with dung and chemical lures.

## Data Availability

All data are stored in archived datasets as per the guidelines of Charles Sturt University and associated funding bodies.

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
