# Peer review of "Contrasting Volatilomes of Livestock Dung Drive Preference of the Dung Beetle Bubas bison (Coleoptera: Scarabaeidae)"

_molecules, 2022, doi:10.3390/molecules27134152_

Round 1
Reviewer 1 Report
This manuscript tested the behavior response of dung beetles to livestock dungs, including horse, sheep and cattle dungs. Furtherly analyzed the VOCs of different dungs. Finally found the horse dung was most chemically diverse and found some components can attract Bubas bison. This experiment designed good and interesting for us to know the seeking response of dung. But there are also some improvements is needed.
1. The food source of different livestock must be clearly, such as pasture and lucerne hay, better complement the Latin name. English name is not formal and also can be confused.
2. In Fig 10, I also confused the vial is under the false floor and does its tight junction with the hole? In the figure showed have some space.
Author Response
Thank you for giving us the opportunity to submit a revised draft of my manuscript titled ‘Contrasting Volatilomes of Livestock Dung Drive Preference of the Dung Beetle Bubas bison (Coleoptera: Scarabaeidae)’. We extend our gratitude for the valuable time and effort you and the reviewers had placed to review this work, and for providing more insightful comments on this work. We have been able to incorporate most of the suggestions provided by the reviewers as explained in this document and changes are highlighted in red. All line numbers and figures are referred to the revised manuscript with tracked changes. Please see the uploaded point by point response to specific issues.

Reviewer 2 Report
The authors Perera et al. prepared a study, which is focusing on the volatile profiling of various host dung species for the Dung beetle Bubas bison and the physiological as well as behavioral readout of the VOCs by the beetle. Both main parts of the study are soundly conducted and support the conclusions drawn. Nevertheless there remain some open points I would like the authors to address or comment on.
Major points:
- Were the beetles in the successive behavioral tests always tested in the same row? If so, would you think that this includes a bias over the course of the whole experiment? I appreciate that you put quite some effort in ruling out any bias here but this could have been one more possibility.
- Did you ever test the dung samples against the empty control to check for putative aversive responses? I had to think of this since the cattle sample performs worse than the control samples.
- Why is the horse-GC/MS-trace in Fig 7 so different from the one shown in Fig 3a? Especially the peaks between 18 and 20 minutes retention time are missing in the pasture fed traces in Fig7. If there is such a variability in the volatilome it would have been great if you generated several per dung type. Or did you check in this?
- Since a SPME can be used with various fibers with varying absorbance parameters, I wonder if you tested several of those before you decided on the one that you used. In addition, it would be nice if you could indicate the used fiber in the Materials and Methods section.
- I was a bit puzzled by your EAG results as I did not really understood them as physiological results in the beginning. Part 2.4. needs some more rewriting in my opinion. Actually I’M not sure if the data shown in Figure 8 and Figure 9 is EAG or behavioral responses. If those are derived from behavioral experiments, I don’t understand the importance of doing the EAGs at all, as they don’t really add anything to the study. But if they are EAG data you would have to make it clearer to the reader. Maybe it is a possibility to add the sample EAG traces from the supplements to the main figure 8
- Still on Figure 8 I would like to hear your opinion on the very variable control response. Do you maybe have an issue with contaminations here?
- More a general note: Did you consider to include a marsupial dung sample as a comparison of this more native substrate of the beetles?
Minor points:
- I would prefer to have the same scaling throughout Figure 1 to better compare the actual readout.
- The text in Figure 6 is slightly quenched and the chemicals are a bit hard to read.
- I would recommend to combine Fig 3,4 and 5 in one as they are all based on the same dataset anyway.
- Some exemplary typos:
o Line 138-139: “ruminat” or “ruminant”?
o Line 161: “lycerne-hay”

Author Response

(The authors gave the same response as above.)
